# Position Feedback-Control of an Electrothermal Microactuator Using Resistivity Self-Sensing Technique

**DOI:** 10.3390/s24113328

**Published:** 2024-05-23

**Authors:** Alongkorn Pimpin, Werayut Srituravanich, Gridsada Phanomchoeng, Nattapol Damrongplasit

**Affiliations:** 1Department of Mechanical Engineering, Faculty of Engineering, Chulalongkorn University, Bangkok 10330, Thailand; 2Micro/Nano Electromechanical Integrated Device Research Unit, Faculty of Engineering, Chulalongkorn University, Bangkok 10330, Thailand

**Keywords:** self sensing, feedback control, resistivity, electrothermal, microactuator, nickel, electrodeposition

## Abstract

The self-sensing technology of microactuators utilizes a smart material to concurrently actuate and sense in a closed-loop control system. This work aimed to develop a position feedback-control system of nickel electrothermal microactuators using a resistivity self-sensing technique. The system utilizes the change in heating/sensing elements’ resistance, due to the Joule heat, as the control parameter. Using this technique, the heating/sensing elements would concurrently sense and actuate in a closed loop control making the structures of microactuators simple. From a series of experiments, the proposed self-sensing feedback control system was successfully demonstrated. The tip’s displacement error was smaller than 3 µm out of the displacement span of 60 µm. In addition, the system was less sensitive to the abrupt temperature change in surroundings as it was able to displace the microactuator’s tip back to the desired position within 5 s, which was much faster than a feed-forward control system.

## 1. Introduction

The development of a micromachined microactuator is very crucial for various emerging applications, including mechanical cell-manipulation such as grasping, handling and releasing [1]. This is because of its perfect size-matching to single cells and capability of generating or measuring microscale motions and forces. Thus, it is straightforward and convenient to adopt microactuators for cell capture, manipulation and characterization.

Among various principles, an electrothermal actuator is attractive for this kind of applications since it provides a small footprint, large displacement, low driving voltage, simple structure, high robustness and ease of fabrication [2,3,4]. The main drawback of the electrothermal actuator is its sensitivity to the temperature of its surroundings. The response of microactuators such as displacement or force would be varied depending on the relative temperature of the actuating element and the surroundings, so displacement or force sensing-system integration is essential to avoid such disturbances for their greater functionality and wider utilization. Furthermore, during the process of grasping, handling or releasing of the object, the displacement or force that the object experiences via the microactuators should be monitored in order to ensure that the object, which is especially tiny in scale, is safety.

Regarding this aspect, the sensing technology of either displacement or force plays a significant role in the development of the microactuator for these modern applications, and various sensing techniques such as piezoresistive [5,6,7,8], capacitive [9,10], thermal [11,12,13] and resistive [14] sensors, integrated with the electrothermal microactuators, have been reported in the literature. However, most devices with elaborate micro-scale sensing elements would require a complex micro-fabrication process. These requirements would lead to a limitation in configuration, dimension and material variation in the devices as the results of design restriction, low adoptability and high cost.

Among these works, it is therefore of interest to use the same actuator structures simultaneously for both actuating and sensing. This technique is called self-sensing technology, which helps to reduce the complexity of the structures of a microactuator. It has been employed in various configurations and applications recently.

In the last few years, our group has developed electrothermal microactuators using an electrodeposition process to form the nickel structures as heating elements [15]. To actuate the system, a potential voltage is applied across the two bond pads, which induces a current through the heating elements at the middle. The current generates Joule heating, and as the temperature of the heating elements rises, they expand. This small elongation of the heating elements is amplified by the actuator’s mechanisms and displaces the actuator tip in the desired direction. A fabrication process using an electrodeposition technique and a configuration of nickel actuators is shown in Figure 1.

In this configuration, the heating element could be used as a sensing element of resistance change as well. For the sensing aspect, two effects could lead to resistance change in the heating elements, namely, temperature change and piezoresistivity. Firstly, the resistivity rises as its temperature increases due to the Joule heating of the applied current. Secondly, with an additional external force, the resistivity at the given current would further change as well, due to the piezoresistive effect. In general, there are two approaches to handle this complexity of the self-sensing technology of electrothermal microactuators.

The first approach is to completely separate the actuating and sensing elements in such a way that the heating is solely applied to the actuating components while the feedback signal was independently examined on the sensing ones. For example, Messenger et al. (2009) utilized polycrystalline silicon as both heating and piezoresistive-sensing elements for a position control of an electrothermal microactuator [6]. To simplify the operation, the driving current was only passed through the heating elements that were absolutely separated from the piezosensitive-sensing elements. While the heating elements were actuated, the microactuator’s mechanisms either elongated or contracted, and the sensing elements only detected the induced strains. Chow and Lai (2009) employed a thermal-based displacement transduction (heat transfer from the actuating elements to sensing element) principle to sense the displacement of electrothermal actuators [11]. The V-shaped actuator was made of electroplated polycrystalline nickel as the heating element. In addition, the sensor was made of polysilicon material and placed underneath the heating elements with an air gap of 1 µm to sense the relative displacement between the heating element and itself.

The second approach is to use the same structures for both heating and sensing elements with additional restraints, for example, the calibration data of effects of each input on the properties change in the output. Ouyang and Zhu (2012) developed the microactuators using n-type (phosphorus) doped single crystalline silicon [7]. To tackle the complex phenomena, a feedback system based on the calibrated relationships between two inputs, namely applied current and external force, and two outputs, namely displacement and electrical resistance, were developed. With this calibration data, the system could generate an updated current to heat and displace the crystalline-silicon structures to the desired position, e.g., a position before external force applied. Recently, Amjadi and Sitti (2018) employed a paper substrate with composite polymer between graphite microparticles and carbon nanotubes as the electrothermally driven microactuators [9]. The heating and sensing were applied simultaneously to the composite polymer film directly. To decouple temperature and piezoresistivity effects on the resistance change, the optimal composition in the form of hybrid polymer films was examined. By finely tuning the charge transport properties of hybrid films, the self-sensing actuators would actively track only the piezoresistivity as the feedback signal since the thermal resistivity of the optimal compositions does not significantly change with the temperature of actuator.

For metal, the change in resistivity due to the piezoresistivity is much smaller than that caused by the temperature change. In the case of the small magnitude of external force in a cell manipulation, which has relatively small mechanical strains, the small resistivity change caused by the piezoresistive effect could be negligible [16,17,18,19]. It would be approximated that the resistance variation in metal is resulted solely from the temperature change; therefore, the self-sensing method using the electrical resistance as a control parameter can be used in dynamic situations when the temperature and the deformation of the actuator are changed at the same time.

Recently, there was some research work demonstrating the resistive self-sensing technique for metal electrothermal microactuators. For example, Tang et al. (2019) used nickel wires to heat the elements and measure its electrical resistance at the same time to develop a closed-loop position control of twisted and coiled actuators [20]. Cao and Dong (2020) employed the resistance change in the heated alloy Bi58/Sn42 to sense the bending curvature of a soft actuator and to develop a closed-loop control system [21]. However, these works did not focus on the cycling operation and its performance when disturbance is applied.

In this work, we have explored the position feedback-control technique by establishing self-sensing systems, utilizing a calibrated relationship between the tip-displacement and thermal-resistance of the hybrid heating/sensing nickel microstructures from our previous work. The developed system was tested to examine the effects of cycling operation and environmental disturbance on the microactuator’s performance. In addition, the fabrication was improved from previous experiments to achieve the uniformity of nickel structures leading to better actuator performance.

## 2. Materials and Methods

### 2.1. Electrothermal Microactuator

The electrothermal microactuator is composed of small and large arms, arranged in a Z-shaped structure as shown in Figure 1. The process flow of the fabrication, solely using the electroplating of nickel structures inside the photoresist mold, is shown in the figure. To improve the uniformity of material properties and dimensions, the pulse electrodeposition of nickel at 67.5 A/dm^2^ in the frequency range of 10–500 Hz (with 50% duty cycle) was employed to construct the microactuator structures. From the examinations, we found that the surface roughness of the pulse-electrodeposition at all conditions was comparable, and decreased from that of the direct-current electrodeposition by about 20%. In addition, the crystalline structure was face-centered cubic, and the lattice constant of crystalline preference orientation (111) was around 3.5 A.

Regarding the operation, when heated, two small arms, or heating/sensing elements, thermally expand, resulting in a motion of the nickel structures. The tip displacement is approximately linearly related to the elongated length of the heating/sensing elements. From the relations of the material expansion and resistivity change due to the heating and cooling, the tip displacement of the microactuator would be a function of the resistivity change due to the rise of temperature as (modified from [15])
(1)$$d=\sqrt{1+{\left(\frac{s}{L}\right)}^{2}}\frac{L\beta }{\alpha }\frac{∆R}{R_{0}}$$
where *d* is the tip displacement, *s* is the distance between small and big arms, *L* is the half-length of small arms, Δ*R* is the change in electrical resistance due to temperature change (Δ*T*), *R*_0_ is the initial electrical resistance, *α* is the temperature coefficient of resistance and *β* is the coefficient of thermal expansion. Equation (1) shows that the displacement would linearly vary with the resistance change in a certain range of temperature; therefore, the resistance change would be a good parameter for the feedback control of the tip displacement.

In this work, the actuator has a shape similar to that in [15]. The small arms are 0.5 mm wide and 10 mm long, and their thickness is 0.2 mm. The gap between the small and big arms is 0.2 mm. The mathematical model is developed to predict the temperature change as a function of the driving current, as shown in Figure 2a,b. The first term on the right side is the amount of Joule heat generated, while the second and third term is an amount of heat conduction and convection, respectively. Figure 2b shows the prediction of the temperature rise for the developed actuator at the applied current of 1–3 A. The resistivity (*ρ*), density and heat capacity (c*_p_*) is 7 × 10^−8^ Ohm·m, 8900 kg/m^3^ and 440 J/kg·K, respectively. Thermal conductivity (*k*) is 95 W/m·K while the free-convection heat transfer coefficient (*h*) is assumed to be 120 W/m^2^·K. The results show that the temperature raise over room temperature (*T* − *T_∞_*) is about 10–50 °C when the current is 1–3 A. In addition, the temperature becomes constant at approximately 10 s after applying the current, and it takes a longer time at a higher current. It should be noted that the resistance change as well as the tip displacement of this microactuator would vary linearly with this temperature rise.

### 2.2. Control System

The mechanical and electrical responses of the microactuator were examined using an experimental setup, as shown in Figure 3a,b. The microactuator was installed in a partially closed box with a glass cover to prevent unrestrained disturbances. The microactuator anchors were fixed on plastic supports placed inside the box. The box was then placed on the microscope stage to simultaneously monitor the displacement of the microactuator through the glass cover as shown in Figure 3a. An electric fan was also installed above the setup to create flow disturbances when needed. It was used to maintain the temperature of circuit boards and all electronic devices at a cool level as well. The electrical resistance of the actuators was examined using Ohm’s law by monitoring the instantaneous potential voltage and current of the heating/sensing elements.

To actuate and monitor the microactuator, the experimental circuit was set up as shown in Figure 3b. The electrical current was passed through the microactuator structure using the current circuit control system. This control system consisted of a power supply, two hall-effect current sensors, and a MOSFET Vishay IRL510 (Vishay Intertechnology, Inc., Malvern, PA, USA) controlled using a signal from the Arduino MEGA 2560 through a 12-bit digital-to-analog converter MCP 4725 (Microchip, Tampa, FL, USA). The wiring diagram is illustrated in Figure 4. The power supply provided 9.7 volts to the circuit, and the Arduino MEGA 2560 controlled the amount of current by adjusting the time interval for triggering the MOSFET and the current amount to the MOSFET through the 12-bit digital-to-analog converter. When the MOSFET was off, there was no current in the circuit, and when the MOSFET was on, the current flowed through the circuit. To measure the current in the circuit, two hall-effect current sensors were set up in a differential-sensor configuration. These hall-effect current sensors were connected in series to the microactuator, but each sensor was connected to a different channel for measuring the differential current. This technique enhanced the sensitivity of the hall-effect current sensors with a noise-canceling technique. The uncertainty of the current sensors was approximately ±2 mA in experiments.

The technique to measure the voltage across the microactuator when the voltage supply from the power source was higher than 5 volts is complicated since the Arduino MEGA 2560 cannot directly measure voltages higher than 5 volts. In addition, the resolution to measure the voltage was low (10 bits). To resolve the problem, the voltage divider circuits were designed and implemented. Resistances of 3000 ohms were used to make the output voltage of the circuit fall within the range of 5 volts. The voltage divider circuit is shown in Figure 4. Additionally, two external 16-bit analog-to-digital converters ADS1115 (Texas Instruments, Dallas, TX, USA) were used to increase the measurement resolution, and the resolution of the designed measurement circuit was approximately ±3 mV. Then, the two analog-to-digital converters were used to simultaneously measure the voltage at the upstream and downstream of the microactuator. The voltage across the microactuator could then be computed by finding the difference voltage between them. Consequently, the instantaneous resistance of the actuator could be computed from the measured current and voltage across the microactuator. With the designed circuit, the system could provide a resistance uncertainty of about 1% (or less than ±2 mΩ).

To independently examine the microactuator tip’s displacement, the photographic technique was employed. Two consecutive images captured from a video recorder connected to a stereo microscope were compared, and the moving distance was measured with a tracking software. The measurement resolution was around 1 µm, while the uncertainty was less than ±6 µm.

## 3. Results and Discussions

### 3.1. Characterization and Control Algorithm

In the system characterization, the direct current at 1.2, 1.6, 2.0, 2.1 and 2.3 A (background current = 0.25 A to detect R_0_) was fed to the microactuators, and the electrical resistance and tip displacement were examined simultaneously. Examples of the experimental results are shown in Figure 5a. The resistance and tip displacement were increased due to the increment in temperature when applying current through a heating/sensing element.

After applying the current, both resistance and displacement were sharply increased and rebounded. At the necking parts of the heating/sensing elements that had a relatively narrow cross-section, the rate of heat generation at these local locations was relatively large. This would result in the locally large thermal-elongation as well as a long tip-displacement. After the heat transfer from those local locations to the entire structures and surroundings gradually occurred, the average temperature as well as the average resistance of the heating/sensing elements reduced temporarily. As a result, the tip displacement of the microactuator rebounded. Beyond this period, the tip displacement slightly fluctuated and then remained unchanged approximately after 10 s from the start, when the heat generation and heat transfer to surroundings were in equilibrium. However, it was observed that the average electrical resistance was still slowly increased, which implied the existence of inertia effects on heat transfer as a result of gradually increasing the average temperature of the microactuator’s structures.

In experiments, at 15 s after applying the current to the microactuators when the change in the tip displacement was less than 1%, the data were collected and plotted. It was found that the relationship between the resistance increment of the microactuator’s heating/sensing element and tip displacement was almost linear as shown in Figure 5b. In the range of the 20-mΩ variation, the tip of the microactuators could travel about 60 µm.

Regarding the results, the feedback control system was developed with the monitoring of the resistance as a control parameter. The drawback of the current system is the need of the individual evaluation of a control function for each microactuator as the characterization results showed that the difference between the resistance change and tip displacement would be possibly significant among tested microactuators from different batches of micro-fabrication. Nevertheless, this issue could be resolved when highly precise fabrication techniques are instead employed in the future.

At this stage, a simple control algorithm was employed. Figure 6 shows a control algorithm that starts with readings of instantaneous voltage and a current of the heating/sensing element every 200 ms. Then, the resistance and subsequently displacement are calculated, and compared with the desired one from the calibrated relationship. If the microactuator has a shorter displacement compared to the desired one, the current with a stepwise increment of 40 mA will be fed to the heating/sensing element. Repeating the methods as mentioned, the current was continuously regulated to achieve the desired displacement.

### 3.2. System Evaluation

There were three consecutive experiments to evaluate the performance of the control system. The first experiment was designed to test a control algorithm, by which the tip of the microactuator was displaced to two locations, i.e., 15 and 25 µm. The second experiment was the monitoring of the continuous step response by comparing the precision between the feedback control and feed-forward (applied constant current) control scheme. The last one aimed to examine the control ability that responded to a sudden change in surrounding.

In the first experiment, the results of the displacement of the tip, which were raised up suddenly and rebounded back to 15 and 25 µm and, after that, held constant, are shown in Figure 7a,b, respectively. The experiments were conducted at the same temperature (room temperature) with those in the characterization tests (as R_0_~140 mΩ). For the 15 µm test, at 5 s after the starting of control, the magnitude of electrical resistance reached the desired value, and afterward stayed almost constant (fluctuation of ±0.5 mΩ). On the other hand, the tip displacement sharply rose almost two times beyond, and then rebounded back to the desired one within 20 s with the position error less than 5 µm. The increment of the tip displacement had a similar trend to the increment of the current that was automatically adjusted. In addition, it was observed that the time interval for the resistance to reach the desired level was longer when the desired tip displacement was 25 µm. The driving current required to maintain the tip displacement of 15 and 25 µm was about 1.5–1.6 and 1.8–1.9 A, respectively. From the mathematical model, the temperature would be about 10 and 20 °C above room temperature for the two cases, respectively.

The increment rate of the driving current would have a large influence on the motion of the controlled microactuators, especially when the rate of heat generation was relatively high. The overshooting of the tip displacement, even larger than that that appeared in the characterization test, would occur. For example, when the driving current was large, the heating elements still had a low temperature due to the inertia effects on the heat transfer. In this situation, the controller detected the low magnitude of electrical resistance. Thus, it raised the driving current more and more as the driving current sharply rose as shown in the figure, resulting in the overshooting of the tip displacement as explained. However, this unintentional motion could be further reduced via an optimization of the control parameters such as the increment rate of the driving current.

Figure 8a,b show the experimental results of the continuous step responses when the desired displacement was temporally increased. The increment of each step was 5 µm in every 10 s. It should be noted that the experiments were conducted at a relatively warmer initial condition than the heating/sensing elements (as R_0_~143–145 mΩ) since the system was operated continuously. Firstly, it was observed that the overshooting of the tip displacement almost disappeared. It suggested that the inertia effects on heat transfer were relatively less when the narrower stepwise increment of the tip displacement as well as resistance was required. With this continuous operation, the current required to maintain the tip displacement at 15 µm was only in the range between 1.2 and 1.3 A. It implied that the temperature of the heating/sensing elements slightly reduced from that in the previous experiment. On average, the feedback control provided a two times smaller error of position than the system driven using a feed-forward control since the system was less sensitive to the unintentional change in surrounding. The position variation in the feedback control was less than 3 µm.

The benefit of utilizing the feedback control is to avoid the effects of the abrupt change in surrounding. As shown in Figure 9a,b, when an electric fan to create a 20-s periodic disturbance was installed, such a disturbance had less effect on the system as the feedback control could shorten the time interval for the tip to move back to the desired position. The heating/sensing elements had the resistance approximately equal to that in the characterizations (as R_0_~140 mΩ). After the fan was turned on (on condition), the tip displacement was suddenly decreased due to the temperature drop. The feedback control responded to this abrupt change, and increased the driving current to heat the microactuators. After 5 s, the tip moved back to the desired level. However, there was no overshooting when the narrow stepwise increment of the tip displacement was required. When the fan was turned off (off condition), the tip displacement was suddenly increased, and rebounded back within 5 s. The driving current required to maintain the desired tip displacement (15 µm) was in the range between 1.4 and 1.8 A that was comparable to the current acquired previously.

On the other hand, with the feed-forward control, the tip displacement was dependent on the flow disturbance. In addition, the tip displacement was gradually increased when time passed by, and the responses of the miroactuators were sensitive to this change. This might be due to the accumulation of heat in the system after the continuous operation.

In summary, this study successfully developed the feedback control of the tip-position for the electrothermal microactuator using the resistivity self-sensing technique. It was found that the feedback control was feasible, provided a smaller positioning error, and made the system less sensitive to the surrounding conditions. However, the non-uniformity of the structure dimensions from the fabrication resulted in the need of an individual characterization of the control function for each microactuator. However, this issue could be resolved using highly precise fabrication techniques. In addition, the control parameters should be further optimized to reduce the unintentional motion of the microactuators.

## 4. Conclusions

This work aimed to develop a position feedback-control of electrothermal microactuators using resistivity self-sensing nickel structures as heating/sensing elements for the first time. The Z-shaped microactuators with the small and big arms were fabricated using the pulse electrodeposition of nickel to improve the uniformity of the material and the actuator’s dimensions. The control systems were relied on to the examine the potential voltage and current across the actuators’ arms, i.e., the heating/sensing elements. It was found that the resistance change in heating/sensing elements had a linear relationship to the tip displacement of microactuators when the fixed current was applied as a static-test manner. Using this relationship as the control function, three tests were consecutively performed. It was found that the response of the system would be dependent to the increment rate of the driving current. If the increment rate of heat generation was large relative to the heat transfer rate of the entire microactuator’s structures and surroundings, the overshooting of the tip motion would occur. In the step-response experiments, the feedback control provided fewer positioning errors compared to that of the feed-forward control, and the error was less than 3 µm. After intervention using flow disturbance, it was observed that the microactuator with the feedback control could respond well and displace the tip back to the desired position within 5 s. The results suggested that the resistivity self-sensing feedback system was feasible and made the actuator responses less sensitive to the abrupt change in surroundings.

## Figures and Tables

**Figure 1 sensors-24-03328-f001:**
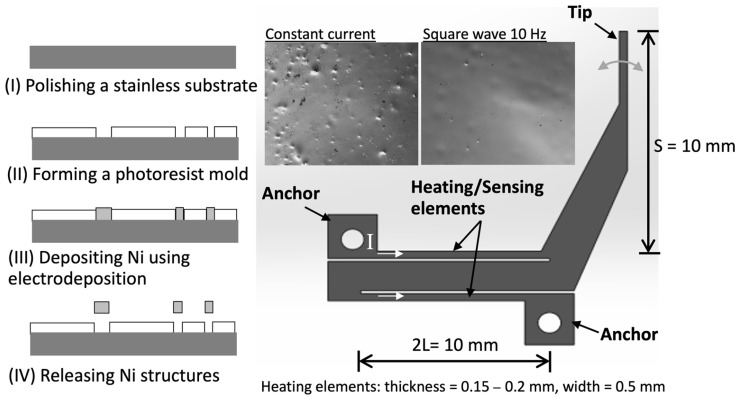
Electrothermal microactuator with dimensions and its fabrication processes using electrodeposition technique.

**Figure 2 sensors-24-03328-f002:**
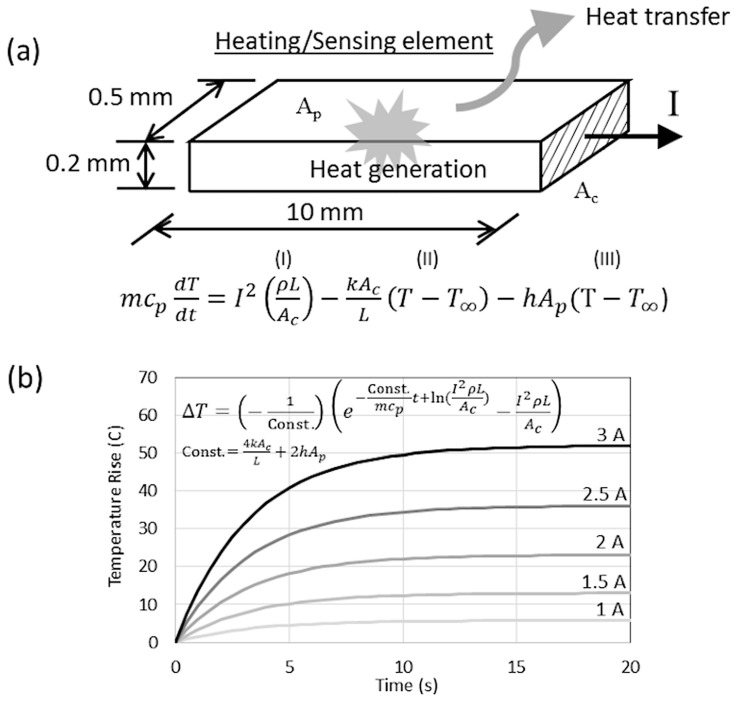
Mathematical model to predict the rate of heat transfer and temperature rise: (**a**) heating/sensing element, and (**b**) computational results.

**Figure 3 sensors-24-03328-f003:**
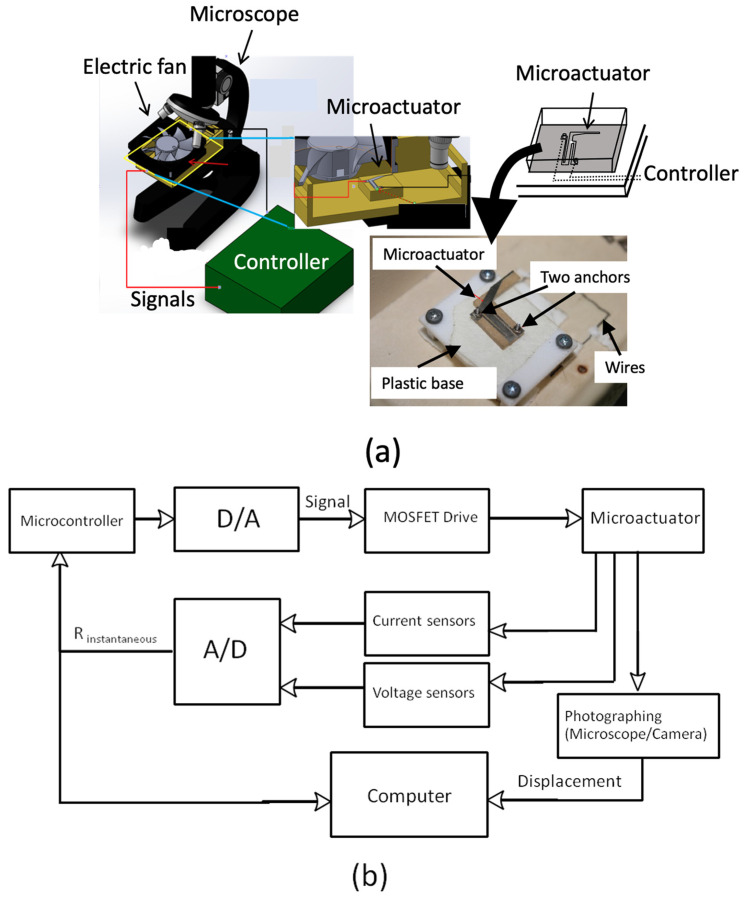
Experimental setups: (**a**) electrothermal microactuator and its installation, and (**b**) control diagram.

**Figure 4 sensors-24-03328-f004:**
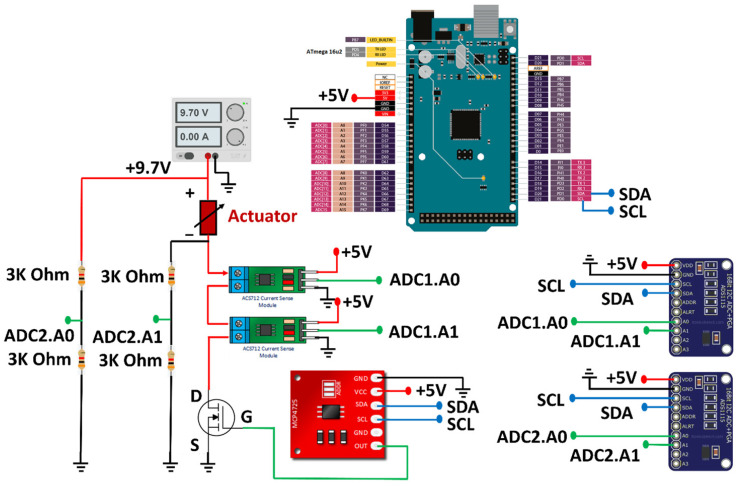
Wiring diagram of sensing and actuating signals from the heating/sensing elements.

**Figure 5 sensors-24-03328-f005:**
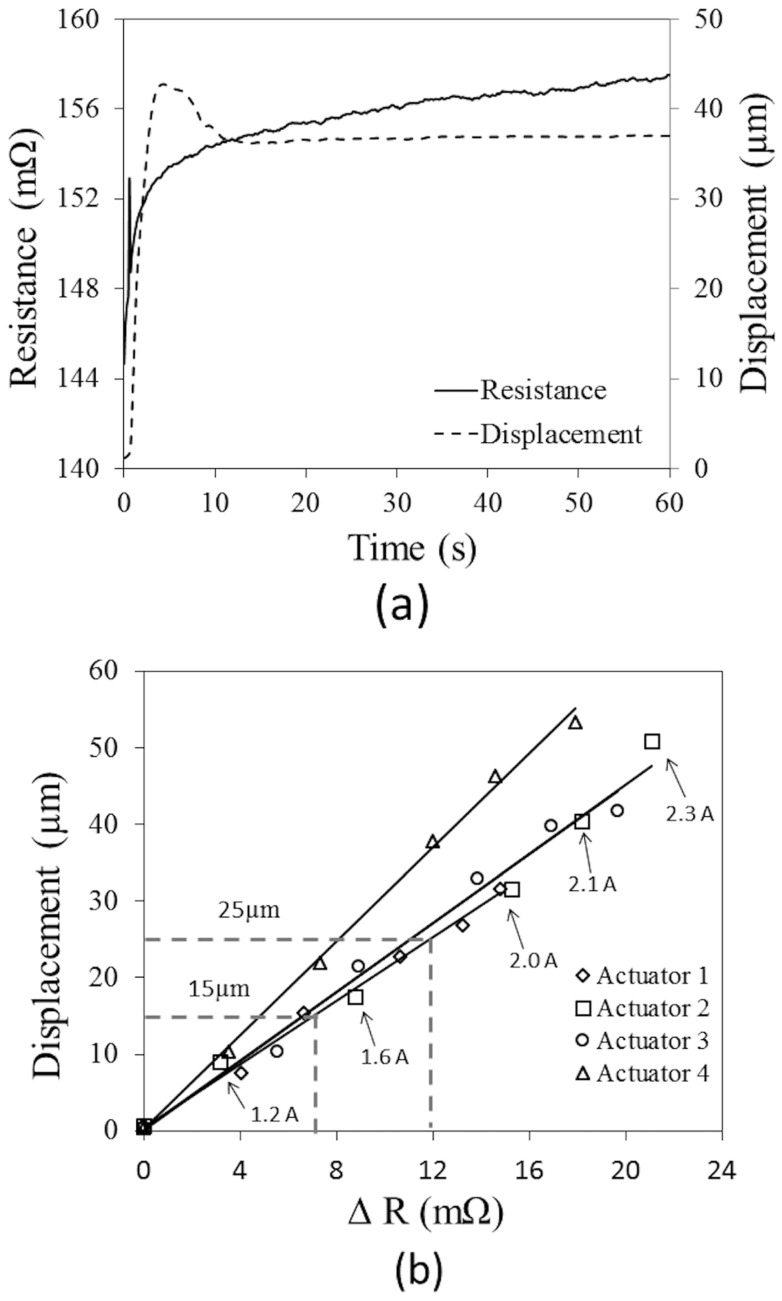
Experimental results to establish the control function: (**a**) instantaneous displacement and electrical resistance, and (**b**) relationship between the tip displacement and the resistance change at 15 s after applying current.

**Figure 6 sensors-24-03328-f006:**
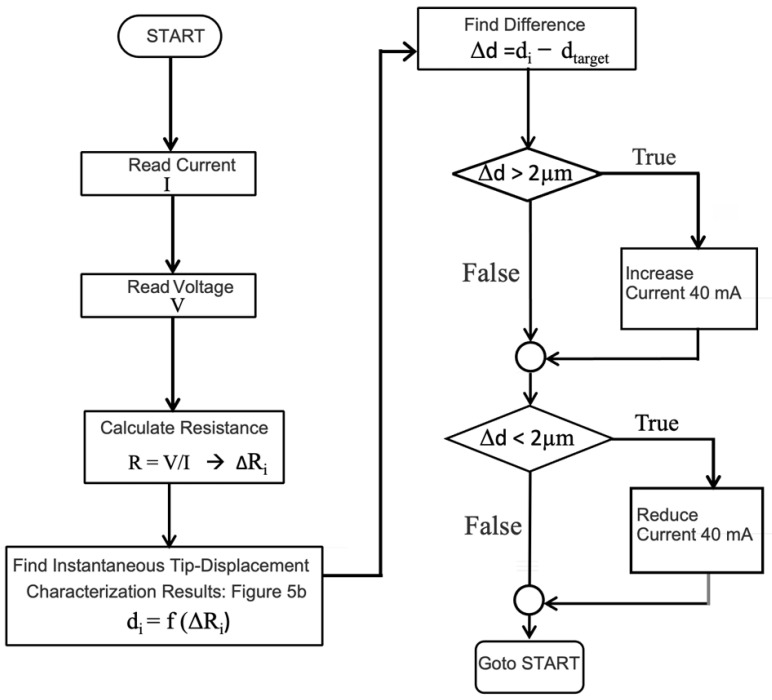
Control scheme and parameters.

**Figure 7 sensors-24-03328-f007:**
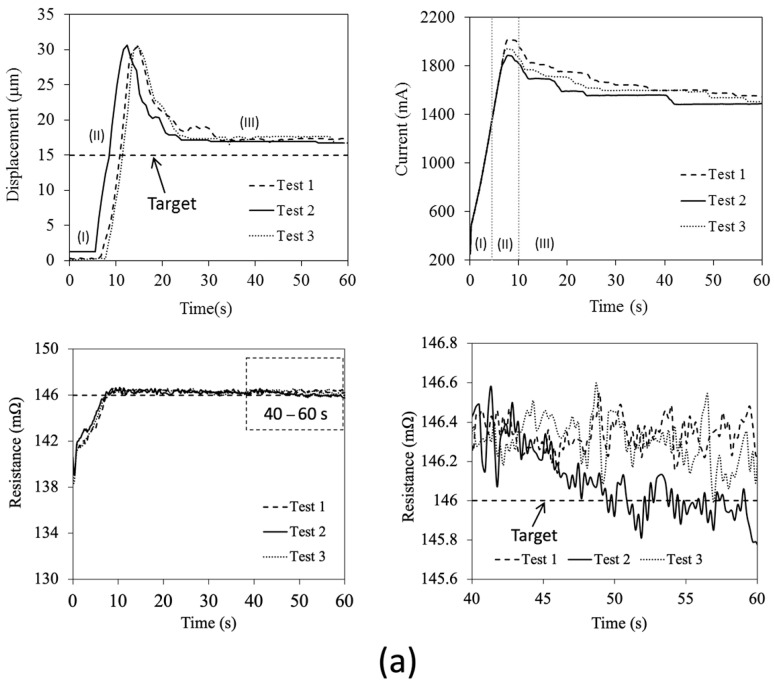
Experimental results of instantaneous tip displacement, driving current and electrical resistance when driven and holding the microactuator’s tip at the distance of (**a**) 15 µm, and (**b**) 25 µm. Region I represents the time interval immediately after the current was applied. Region II represents the time interval when the tip suddenly moved, and Region III represents the time interval when the tip was held constant.

**Figure 8 sensors-24-03328-f008:**
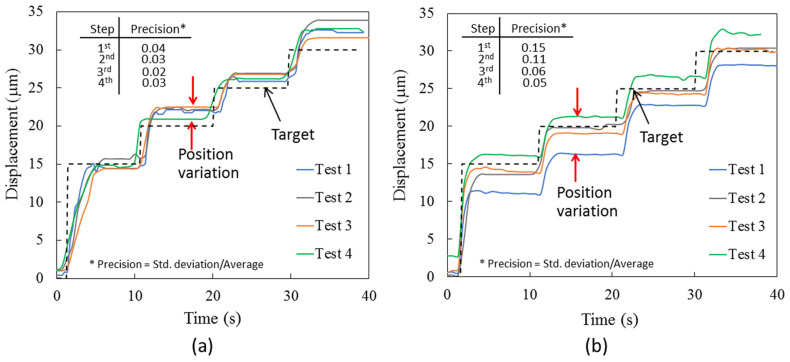
Experimental results of step responses from 15 to 30 µm for (**a**) feedback, and (**b**) feed-forward control.

**Figure 9 sensors-24-03328-f009:**
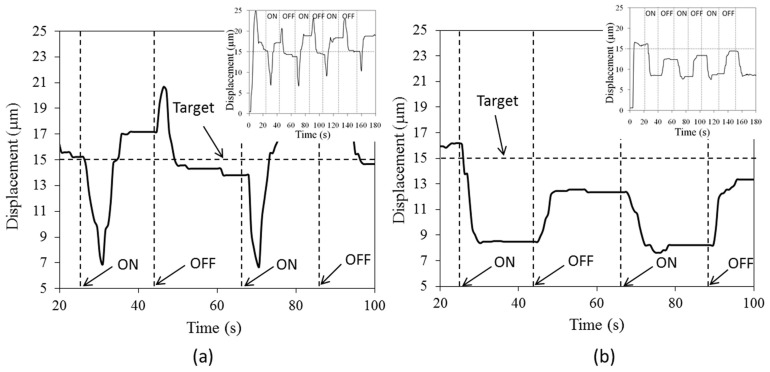
Experimental results when applying surrounding disturbance every 20 s for (**a**) feedback, and (**b**) feed-forward control.

## Data Availability

Any inquiry can be directly sent to the corresponding author.

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
