# Peer review of "Position Feedback-Control of an Electrothermal Microactuator Using Resistivity Self-Sensing Technique"

_sensors, 2024, doi:10.3390/s24113328_

Round 1
Reviewer 1 Report
Comments and Suggestions for Authors
This work presents details on a feedback system for self-sensing of electrical resistance and sub-sequent position control of a micro-electrothermal actuator. The work is well done, and details are presented in a clear and straightforward way. Some comments below:
1. The control scheme and electronics shown in Fig. 3 And Fig. 4 are valid and very detailed. Are you controlling the current flow to the actuator by biasing the MOSFET near the sub-threshold region? Isn’t this less ideal and may cause overheating of the MOSFET? Or are you doing pulse width modulation to control the current (but there does not seem to be a low pass filter)? Another method to control this is using something similar to a buck converter, using PWM to control the current flow and DC voltage to the load, which would be the actuator.
2. Could you explain in more detail why using two Hall-effect sensors helps you do differential current sensing? The current is flowing in the same direction, so how does this help with noise reduction?
3. Can you also explain how you accomplish position sensing with the image capturing system? Are you adjusting the focal plane to get the data? Or have you considered using laser Doppler vibrometer techniques to do this? However, since you device is sealed, the laser method will run into limits I believe (but might be more accurate).
4. Is the continuous rise in resistance due to the fact that the actuator is sealed, and the heat cannot escape very well? What do you mean by temperature inertia effects?
5. You can also cite some other MEMS electrothermal work that relate to feedback control with position sensing schemes as below:
[1] V. F.-G. Tseng, et al., “Resonant inductive coupling-based piston position sensing mechanism for large vertical displacement micromirrors,” JMEMS, vol. 25, no. 1, 2015
[2] Y. Tang, et al., “A 2-AXIS SI/AL Bimorph-Based Electrothermal Micromirror Integrated with Piezoresistors for High Resolution Position Sensing,” 2024 IEEE 37th International Conference on Micro Electro Mechanical Systems (MEMS).
Author Response
We would like to thank the reviewer for your kind consideration. The attached file is the responses to your questions/comments.

Reviewer 2 Report
Comments and Suggestions for Authors
Please see the attachment

Author Response

(The authors gave the same response as above.)
